Finite element analysis of root canal filling materials in retained primary molars with congenital tooth agenesis

Karagöz Doğan Gizem dtgizemkaragoz@gmail.com 1
Yavuz Yelda Polat 2
Karaağaç Eskibağlar Büşra 3
1 Faculty of Dentistry, Iğdır University , Iğdır , Turkey
2 Faculty of Dentistry, Dicle University , Diyarbakır , Turkey
3 Faculty of Dentistry, Fırat University , Elazığ , Turkey
Sergi Consolato
Electronic publication date: 2025 Oct 16
Publication date: 2025
Volume: 13
Electronic Location ID: e20206
Received 2025 Mar 25; Accepted 2025 Sep 17
Copyright: ©2025 Karagöz Doğan et al.
Copyright year: 2025
Copyright holder: Karagöz Doğan et al.
License: This is an open access article distributed under the terms of the Creative Commons Attribution License, which permits unrestricted use, distribution, reproduction and adaptation in any medium and for any purpose provided that it is properly attributed. For attribution, the original author(s), title, publication source (PeerJ) and either DOI or URL of the article must be cited.
License URL: https://creativecommons.org/licenses/by/4.0/

Keywords: Finite element analysis, Primary teeth, Mineral trioxide aggregate, Biodentine

Funding: The authors received no funding for this work.

==============================
Background

The selection of an appropriate filling material in root canal treatments of primary molars is crucial for long-term success. However, evaluating the biomechanical behavior of these materials under in vivo conditions remains challenging. This study aimed to investigate the effect of different root canal filling materials on the fracture resistance of the mandibular second primary molar by analyzing stress distributions and failure risk using finite element analysis (FEA) models.

Methods

A mandibular second primary molar extracted for orthodontic reasons was used in the study. The tooth was scanned using micro-computed tomography (micro-CT) to obtain original DICOM data, which were imported into Geomagic + SpaceClaim 2023R2 to create a solid model. A total force of 330 N was applied to three points on the occlusal surface of each model. The analysis was conducted using Ansys mesh and evaluated with Hyperview 2024. Maximum von Mises (vM) stress values were used to assess stress distribution.

Results

The highest vM stress in the remaining dentin was observed in the gutta-percha (GP) + AH Plus model (148.5 MPa), followed by mineral trioxide aggregate (MTA) (127.24 MPa), Biodentine (125.65 MPa), and GP + BioRoot RCS (118.37 MPa). Stress concentrations were primarily located in the pericervical region. The GP + AH Plus group showed the highest stress, while the GP + BioRoot RCS group showed the lowest. Among contemporary filling materials, GP + BioRoot RCS demonstrated the lowest dentin stress, suggesting it may offer better root fracture resistance. However, generalizing these findings is difficult due to limited data on primary teeth in the literature.

Conclusion

The study suggests that GP + BioRoot RCS may be a more promising filling material for enhancing root fracture resistance in primary molars. Further research is needed to validate these findings in clinical settings.

Introduction

Primary teeth can persist firstly due to the congenital absence of the underlying permanent tooth, and secondarily due to being impacted or migrating within the bone. It is reported that the most commonly seen persistent primary tooth in the mouth is the mandibular second primary molar, followed by the maxillary primary canine. When a persistent primary tooth has a good crown, root, and supportive alveolar bone structure, it can serve an adult individual for many years (Aslan, Akarslan & Uzuner, 2013). Since persistent primary teeth do not shed at the physiological time they should and remain in the arch for many years, they should be considered like permanent teeth. Materials used in root canal treatments of permanent teeth should be employed to maintain these teeth in a healthy and functional state in the mouth for as long as possible (Sogukpinar & Arikan, 2021).

Root canal treatment of primary teeth without a permanent tooth germ underneath is performed with gutta-percha (GP) and resin-based root canal sealer in clinical routine. However, due to irregularities in root canal anatomy compared to permanent teeth and thinner dentin structure than permanent tooth dentin, difficulties arise during treatment (Sogukpinar & Arikan, 2021; Tunc & Bayrak, 2010).

Furthermore, in vitro studies have shown that gutta-percha, commonly used as a root canal filling material, exhibits a certain level of vulnerability to fluid penetration and bacterial leakage, with coronal microleakage being a particularly critical concern. This has led to the idea of using more contemporary bioactive materials, such as calcium silicate-based materials like mineral trioxide aggregate (MTA) and Biodentine, for canal filling, and various studies have been conducted on this (Tunc & Bayrak, 2010; Gul et al., 2018).

MTA is a bioactive material proven to have excellent sealing capability and stimulate hard tissue healing when used in lateral root perforations, internal resorption, furcation perforations, direct and indirect pulp capping, pulpotomy, and root end filling. Biodentine, another member of the bioactive material family, was formulated based on MTA’s disadvantages, such as long setting time and risk of discoloration, and is claimed to have better physical and biological properties than MTA (Zafar, Jamal & Ghafoor, 2020). In the literature, there are studies and case reports evaluating the effect of root canal fillings with GP + AH Plus, MTA, and Biodentine on the biomechanics of the root through clinical-radiographic observations or in vitro using universal testing machines (Gul et al., 2018). However, no study has been found that compares the stress distributions generated by MTA, Biodentine, GP + AH Plus, and their hybrid applications in mandibular second primary molar models using finite element analysis (FEA).

FEA is a computer-based measurement technique used to examine the biomechanical processes of teeth. Due to its ability to simulate the biomechanical behavior of complex anatomical structures, FEA has emerged as a valuable tool, particularly in pediatric endodontics. FEA not only reveals stress distributions and areas of high stress concentration within dental tissues, but also aids in understanding the potential mechanical processes occurring during loading periods. Since direct measurement of mechanical stresses in primary teeth under in vivo conditions is limited by ethical and technical constraints, finite element analysis offers an effective solution to overcome this challenge. FEA enables the highly accurate assessment of stress distributions in dental tissues and allows for the comparison of the mechanical behavior of different materials under load without causing damage to biological tissues. Moreover, FEA has been proven to be a more adaptable, accurate, practical, and time-efficient method compared to other experimental analysis techniques (Wang et al., 2020; Sengul, Gurbuz & Sengul, 2014; Atif et al., 2024).

Various in vivo and in vitro studies in the literature have investigated the biomechanical effects of materials such as MTA, Biodentine, and AH Plus (Karobari et al., 2023; Kaup, Schäfer & Dammaschke, 2015). However, no studies have been found that comparatively evaluate these materials in primary molars using a FEA model. This highlights a significant gap in understanding the mechanical performance of root canal filling materials used to maintain the long-term functionality of these teeth. The aim of this study is to address this gap by investigating the effects of current root canal filling materials on the stress distribution in the root of the mandibular second primary molar using FEA.

Materials & Methods

Ethical approval was obtained from the Firat University Non-Interventional Research Ethics Committee (session number 2022/10-14). Written informed consent was obtained from the parent or legal guardian of the patient for the tooth used in this study. A three-dimensional (3D) model of a second primary molar, extracted for orthodontic purposes and free from decay or fractures, was generated. This tooth was imaged via micro-computed tomography (micro-CT) scanning to acquire the initial DICOM dataset. The first step in FEA is modeling. The quality of the analysis depends on the accuracy of the 3D model (Levrini, Merlo & Paracchini, 2007). The original DICOM data was transferred to Geomagic + Spaceclaim 2023R2, where errors, gaps, and small surfaces in the data were repaired. A solid model was created, and final adjustments were made in Spaceclaim 2023R2 to make the model suitable for analysis. In this study, the tooth was modeled by considering essential morphological structures such as enamel, dentin, pulp, and periodontal ligament. The prepared model was imported into the Ansys software (Ansys Mechanical 2023R2), where a mesh (element) solution network was created (Fig. 1). The number of nodes and elements in the main model are shown in Table 1.

Figure 1 3D finite element mesh of the mandibular second primary molar model.

Table 1 The number of elements and nodes of dental materials, teeth, and periodontal structures.

Dental structure/material	Node	Element	
Enamel	49,051	28,453	
Dentin	109,881	69,824	
Pulp	31,917	17,658	
Periodontal ligament	43,240	21,711	
Cancellous bone	91,161	60,584	
Cortical bone	24,276	11,822	
Glass Ionomer Cement	11,285	7,301	
Composite	16,828	10,743	

Then, four different analysis models were created from the main model:

• Model 1: The root canal is filled entirely with MTA.

• Model 2: The root canal is filled entirely with Biodentine.

• Model 3: The root canal is filled with AH Plus + GP.

• Model 4: The root canal is filled with Bioroot RCS + GP.

All materials were considered to exhibit linear, homogeneous, and isotropic properties, a simplification commonly adopted in finite element analysis to streamline the computational process and ensure numerical stability of the model. Moreover, the majority of similar FEA studies in the fields of endodontics and restorative dentistry employ these same modeling assumptions to enhance comparability across studies. A total of 228,096 elements and 377,639 nodes were used in the models. The material properties (modulus of elasticity and Poisson’s ratio) defining each model and treatment material were obtained from the manufacturer. Information on the modulus of elasticity and Poisson’s ratio is provided in Table 2.

Table 2 The elasticity modulus and Poisson’s ratios of dental materials, teeth, and periodontal structures.

Dental structure/material	Modulus of elasticity (E) (GPa)	Poisson ratio	
Enamel	41	0.31	
Dentin	18.6	0.31	
Pulp	0.003	0.45	
Periodontal ligament	0.000689	0.45	
Cancellous bone	1.37	0.3	
Cortical bone	13.7	0.3	
MTA	11.76	0.31	
Biodentine	22	0.33	
Gutta percha	0.14	0.45	
AH Plus	0.3	0.3	
Bioroot RCS	3.2	0.33	
Composite	12.5	0.3	
Glass Ionomer Cement	20.1	0.3	

The tooth root was constrained in all directions (x, y, z) as a boundary condition. According to a study cited in the literature, the bite forces on primary teeth range from 161 to 330 N (Rentes, Gaviao & Amaral, 2002). During the analysis, the researchers applied a total force of 330 N to three distinct points on the occlusal surface of the model. These points were identified as the central fossa, the mesiobuccal cusp tip, and the distobuccal cusp tip, based on recent FEA studies conducted on primary teeth (Fig. 2) (Guler, 2023). For angled loading, the force was directed from lingual to buccal at a 45° inclination relative to the tooth’s long axis.

Figure 2 Distribution and direction of applied occlusal forces on the model.

Static and linear analyses were conducted to comparatively evaluate the stress distributions generated by different root canal filling materials under controlled conditions. Although nonlinear or dynamic analyses may reflect physiological loading more realistically, such approaches require complex material definitions, loading conditions, and solution criteria—particularly in structures like primary teeth, for which mechanical property data remain limited. Results were evaluated on the Ansys mesh using the Hyperview 2024 software, with visualizations for the study incorporated via this program. Forces were quantified based on peak von Mises (Vm) stress values in MPa. Stress distribution across the models was depicted with a color gradient, ranging from red to blue, indicating decreasing values. Dark blue highlights regions of minimal von Mises stress, while red indicates areas of peak von Mises stress. The area displaying the highest stress level was identified as having the greatest likelihood of failure, guiding further assessments.

Results

In this study, the mechanical properties of materials used in the obturation of a mandibular second primary molar without an underlying permanent tooth germ were examined iotrthe application of forces to the models of mandibular second primary molars filled with MTA, Biodentine, GP + AH Plus, and GP + Bioroot RCS, the highest vM stress value was observed in the enamel tissue of the tooth. Among the enamel tissues, the highest vM stress value was found in the model filled with Biodentine (337.75 MPa), followed by GP + AH Plus (300.9 MPa), MTA (239.71 MPa), and GP + Bioroot RCS (239.4 MPa) models, respectively (Fig. 3).

Figure 3 Maximum von Mises stress distribution in enamel for each canal filling material.

Within the remaining dentin tissue of the models, the highest vM stress was observed in the GP + AH Plus model (148.5 MPa), followed by MTA (127.24 MPa), Biodentine (125.65 MPa), and GP + Bioroot RCS (118.37 MPa) models (Table 3). The maximum vM stress concentrations showed a homogeneous distribution across different models. While the stress concentrations were higher in the pericervical area of the models, it was determined that the vM stresses occurring on the distal side of the root were higher than in other regions of the tooth (Fig. 4).

Table 3 Maximum von Mises stress levels occurring in enamel and dentin in the models (MPa).

	Enamel (MPa)	Dentin (MPa)	
Model 1 (MTA)	239.71	127.24	
Model 2 (Biodentine)	337.75	125.65	
Model 3 (GP + AH Plus)	300.9	148.5	
Model 4 (GP + Bioroot RCS)	239.4	118.37	

Figure 4 Maximum von Mises stress distribution in dentin for each root canal filling configuration.

Discussion

Congenital tooth agenesis is one of the most common clinical problems, especially in children, and is typically observed in third molars, followed by second premolars and maxillary lateral incisors. It is more common in permanent teeth compared to primary teeth. The etiology is not fully understood, though genetic and environmental factors are thought to play a role (Aktan et al., 2012). In a study conducted in Turkey’s Aegean Region, it was found that congenital permanent tooth agenesis in children most frequently occurs in the mandibular second premolar teeth (Candan, Kipcak & Evcil, 2015). This finding aligns with the results of other studies in the literature (Sheikhi, Sadeghi & Ghorbanizadeh, 2012). Persistent primary teeth without an underlying permanent tooth germ can continue to function for many years if they have a healthy crown, root, and supportive alveolar bone structure, making them clinically significant (Tunc & Bayrak, 2010). Considering the literature, our study examined the retained mandibular second primary molar, which is often observed in cases of mandibular permanent second premolar agenesis. If there are signs of infection in the retained primary molar tooth, root canal treatment should be performed. However, root canal treatment is contraindicated in cases where the tooth cannot be restored, severe bone resorption is present, or there is perforation or internal/external root resorption (Ansari & Mirkarimi, 2008).

The preferred treatment method for necrotic pulp in persistent primary teeth is usually traditional root canal treatment using gutta-percha and root canal sealers. However, the anatomical structure of primary teeth differs from permanent teeth; their dilacerated and brittle root morphology complicates the use of standard instruments and root filling techniques (Ulusoy & Cehreli, 2017). While biocompatibility of root canal filling materials is a fundamental requirement for a successful canal treatment, they must also possess adequate compressive strength and hardness to withstand occlusal forces (Basturk et al., 2013). Recent studies suggest the use of MTA in root canal treatment of retained primary molars, in line with the view that gutta-percha may not provide adequate apical sealing in the narrow root canals of primary teeth. MTA is preferred due to its biocompatibility, apical sealing ability, and its support for regeneration in periradicular tissues (Tunc & Bayrak, 2010).

In our study, based on this literature, we examined stress distribution in root canal-treated retained primary molars using different root canal filling materials and techniques: traditional root canal treatment with AH Plus and gutta-percha, root canal treatment with MTA and Biodentine, and a recently introduced calcium silicate-based bioceramic root canal sealer, BioRoot RCS, combined with gutta-percha, using the FEA.

FEA enables numerical predictions regarding the impacts of novel materials or treatments on anatomical tissues, allowing for potential refinements prior to clinical trials. Additionally, FEA is extensively utilized and effectively implemented across disciplines like engineering, bioengineering, and dentistry. In contrast, clinical or experimental studies may be influenced by various factors, including variations in dental anatomy, equipment calibration inconsistencies, and potential author bias (Sengul, Gurbuz & Sengul, 2014; Ordinola-Zapata et al., 2022). Furthermore, FEA studies can only demonstrate stress distributions, and these findings should not be directly equated with fracture resistance without validation through in vivo and in vitro testing.

FEA is a valid method for addressing mechanical performance and interpreting the mechanisms of experimental results in a realistic manner (Lin et al., 2022). Accurately representing the mechanical properties of each material, given their unique mechanical characteristics, is an important factor to consider in FEA analyses. Additionally, in FEA studies, material-related factors such as Poisson’s ratio, Young’s modulus, and the density of each material should also be taken into account (Poiate et al., 2009).

Von Mises stress values are frequently employed in FEA simulations to assess stress distributions induced by applied forces. Existing research highlights that the distribution of maximum stress can vary considerably depending on the direction and location of force application. This underscores the importance of precisely defining both the direction and the area of applied force in FEA simulations to accurately interpret stress patterns (Guler, 2023). In the present study, the stress distribution and analysis resulting from occlusal forces on various root canal filling materials and techniques applied to retained primary molar teeth were evaluated using FEA.

FEA models can be constructed in either 2D or 3D formats; however, 2D models may be insufficient in capturing strain and stress accurately, which can lead to errors in the results due to artificial constraints. As a result, utilizing 3D models for the analysis of biological or biocompatible structures yields more realistic and dependable outcomes. Three-dimensional models provide a more precise representation of complex structural behaviors, simulate natural load and stress distributions more effectively, and offer more comprehensive and detailed insights for biomechanical assessments. Drawing from this understanding in the literature, the FEA model of the retained primary molar in our study was developed and analyzed in 3D (Atif et al., 2024).

Most recent FEA studies utilize computed tomography (CT), cone beam CT, or micro-CT (Lin et al., 2022). Micro-CT allows for the creation of highly accurate and detailed FEA models of small objects, such as teeth, bones, implants, and restorations. This enables the simulation of different treatment methods’ effects on stress distribution in a way that closely resembles real life. Detailed models obtained through micro-CT are particularly useful for analyzing stress distribution on the root canal wall, especially in teeth that have undergone endodontic treatment or contain posts made from different materials. Additionally, data obtained from micro-CT enable broader stress analyses by allowing the formation of various experimental groups (González-Lluch et al., 2014). In our research, micro-CT was employed to generate the three-dimensional solid model of the retained primary tooth. A prior investigation demonstrated that masticatory forces in primary teeth range from 161 to 330 N (Rentes, Gaviao & Amaral, 2002). Additionally, another study applied a force of 330N both vertically and at oblique angles to each model, simulating the maximal occlusal and lateral chewing conditions (Waly et al., 2021).

Owais, Shaweesh & SAbu Alhaija (2013) reported that the maximum chewing force during the early primary dentition period was 176 N, while Abu-Alhaija, Owais & Obaid (2018) reported this value as 197 N. Based on data in the literature, a total of 330 N in vertical and oblique forces was applied at three points on the occlusal surface of the four different models created in our study, thus simulating the chewing forces in primary teeth.

An optimal root canal filling material should not only bond ideally to root canal dentin but also strengthen the remaining tooth structure to increase the long-term success of endodontically treated teeth. In study examining the effect of root canal sealers on fracture resistance of roots, most studies suggested that root canal sealers could increase the fracture resistance of the root (Uzunoglu-Ozyurek, Kucukkaya & Karahan, 2018).

BioRoot RCS, a root canal sealer developed in recent years, is a calcium silicate-based material primarily consisting of tricalcium silicate and zirconium oxide powder, which must be mixed with a liquid containing calcium chloride. Recent studies have indicated that the addition of zirconium oxide to calcium-silicate cements can enhance their biocompatibility and odontogenic potential, supporting their use as reliable materials in vital pulp therapy—particularly for primary teeth. Over the past decade, there has been a significant focus on both conventional and resin-modified calcium-silicate cements. Numerous studies have shown that these materials possess favorable physical, chemical, mechanical, and biological properties in both in vitro settings and clinical applications (Gandolfi et al., 2015; Abedi-Amin et al., 2017; Chen et al., 2009). BioRoot RCS exhibits prolonged calcium ion release and sustained alkalinity after setting, which contribute to its strong antimicrobial properties and low cytotoxicity, thereby supporting both endodontic and periodontal tissue regeneration. Its clinical effectiveness is further enhanced by its ability to provide a reliable seal in moist environments, facilitated by mineralization and apatite formation at the dentin–sealer interface (Srivastava et al., 2020; Camilleri, 2015). On the other hand, AH Plus (Dentsply DeTrey GmbH, Constanz, Germany), an epoxy resin-based sealer, is frequently used in root canal treatments due to its strong bonding to dentin, low solubility, resistance to degradation, and satisfactory dimensional stability compared to other root canal sealers (Zhou et al., 2013). Moreover, AH Plus’s monoblock effect can potentially help reduce harmful stresses in the root (Belli et al., 2011). In endodontics, the monoblock effect refers to the formation of a single homogeneous structure through the chemical or mechanical integration of obturation materials (such as sealer and root canal filling material) with the canal dentin. This unified structure minimizes the formation of voids (microleakage) within the canal and ensures that applied stress (e.g., masticatory forces) is transferred more evenly and efficiently to the canal walls (Baghdadi et al., 2021). Moreover, BioRoot RCS, a root canal filling material based on calcium silicate and possessing a higher elastic modulus compared to AH Plus, is capable of efficiently distributing stresses throughout its internal structure. This leads to a more uniform stress distribution in teeth treated with BioRoot RCS, enabling the material to closely resemble the properties of dentin tissue (Belli, Eraslan & Eskitascioglu, 2016). Furthermore, as reported by Karobari et al. (2023) BioRoot RCS exhibits superior push-out bond strength and deeper dentinal tubule penetration compared to epoxy resin-based sealers, which suggests improved sealing ability and reduced microleakage. These properties may contribute to the material’s long-term clinical success, particularly in pediatric patients where maintaining primary teeth over extended periods is critical.

In this study, we simulated retained primary molars filled with two different root canal materials: the epoxy resin-based AH Plus combined with gutta-percha, and the recently developed calcium silicate-based bioceramic sealer BioRoot RCS with gutta-percha. The stress distribution and analysis of these materials were conducted and compared. In another study, the distribution of stress under both vertical and oblique forces was assessed in permanent mandibular premolars filled with either BioRoot RCS or AH Plus, with or without the addition of gutta-percha. To investigate the impact of various root canal filling materials and techniques on von Mises stress in root dentin, eight distinct scenarios were simulated. These simulations included single-cone techniques with gutta-percha as well as bulk-fill techniques utilizing only sealer, all tested under different force conditions. The findings revealed that the AH Plus model exhibited higher maximum von Mises stress compared to the BioRoot RCS model, indicating that AH Plus’s relatively low modulus of elasticity failed to adequately support the structure, thereby causing increased stress in the surrounding dentin (Smran et al., 2024). In alignment with the literature, in our study, the highest von Mises (vM) stress value within the remaining dentin tissue was measured in the gutta-percha + AH Plus group (148.5 MPa), while the lowest vM stress value was observed in the gutta-percha + BioRoot RCS group (118.37 MPa). Maximum vM stress concentrations displayed a similar, homogeneous distribution across different models.

A study conducted by Brito-Ju’nior et al. (2015) found that the peak von Mises stress (Vm) values in root canals filled with AH Plus were recorded in the coronal third, whereas the lowest Vm values were observed in the apical third of the root canal. Consistent with the literature, in our study, it was found that stress concentrations were focused in the pericervical area of the models, and the vM stresses on the distal side of the root were higher compared to other regions of the tooth.

MTA is known for its superior biocompatibility, sealing ability, regeneration potential, and antibacterial properties, and it has no cytotoxic effects on tissues (Jamshidi et al., 2018). However, to address some disadvantages of MTA, such as its difficulty of application, high cost, long setting time, and discoloration potential, Biodentine has been introduced in recent years, especially as an apical plug material (Topçuoglu et al., 2015). In a study conducted by Eram et al., the fracture resistance of immature teeth using MTA, Biodentine, and Bioaggregate as apical plugs and root canal filling materials in an immature maxillary central incisor was compared using FEA. The study reported that the highest stress concentration in all models was located in the cervical region. Additionally, when the entire root canal system was filled, the lowest stress distribution and level on the dentin surface were obtained with Biodentine (Eram et al., 2020). Similarly, in our study, it was observed that the vM stress value within the remaining dentin tissue was higher in MTA than in Biodentine, with the stress mainly concentrated in the pericervical area of the models. In another study investigating the effects of root canal treatment using MTA and gutta-percha on stress distribution in teeth with internal root resorption, it was shown that MTA reduced stress concentration in resorption areas, while gutta-percha transmitted stress to dentin, increasing the risk of fracture. Based on these findings, the clinical use of MTA alone or in combination with gutta-percha is recommended for teeth with internal root resorption (Aslan, Üstün & Esim, 2019). In a separate study examining stress distribution following the repair of various iatrogenic root perforations in mandibular molars with Biodentine and MTA, it was found that models with Biodentine showed lower peak von Mises stress values compared to those repaired with MTA (Aslan et al., 2021). Similarly, in our study, the highest vM stress value within the remaining dentin tissue was measured in the gutta-percha + AH Plus group, followed by MTA, Biodentine, and gutta-percha + BioRoot RCS groups.

Our study examined root canal filling materials such as MTA, Biodentine, AH Plus with gutta-percha, and BioRoot RCS with gutta-percha. However, the limited data in the literature regarding the use of these materials in primary teeth resulted in some differences and limited comparability of the vM stress findings with previous studies.

Limitations

• This study is the first to investigate the effects of current materials used in root canal treatment of retained primary teeth on root dentin using FEA. Due to the limited data in the literature, generalizing the results is challenging.

• FEA studies can only demonstrate stress distributions, and these findings cannot be directly equated with fracture resistance without validation through in vivo and in vitro testing. The primary aim of this study was not to directly measure fracture resistance, but rather to draw inferences regarding potential fracture risk based on stress patterns. Although there are studies in the literature in which predictions of fracture risk are supported by FEA, the findings of this study are subject to important limitations in terms of their generalizability to clinical applications.

• The study focused on specific root canal materials, such as MTA, Biodentine, AH Plus with gutta-percha, and BioRoot RCS with gutta-percha, providing a limited perspective by not evaluating other biocompatible materials.

• Only the stress distribution from vertical and oblique forces was examined, while other occlusal forces and chewing movements were not considered, partially limiting the findings. However, the use of a single representative tooth model limits the generalizability of the findings; incorporating multiple samples or employing parametric models is necessary to account for anatomical variations.

• In our study, we assumed that the root canal fillings were ideally and completely filled; therefore, voids or defects commonly encountered in clinical practice were not analyzed. Additionally, aging factors such as thermal cycling or microleakage were not included in the analysis, limiting the direct translation of our results into long-term clinical applications. Finally, our investigation focused solely on mechanical stress analysis without assessing the biological effects of materials (e.g., bioactivity, remineralization potential), thus restricting the clinical interpretability of our findings.

Conclusions

This study is the first to examine the effects of contemporary root canal filling materials on dentin in persistent primary teeth using the FEA. Different filling materials, including MTA, Biodentine, AH Plus with gutta-percha, and BioRoot RCS with gutta-percha, were compared, and vM stress distribution was analyzed. The highest vM stress was measured in the GP + AH group, followed by MTA, Biodentine, and GP + BioRoot RCS groups, with stress concentrated in the pericervical area of the models. However, due to the limited data in the literature on primary teeth, generalizing these results is difficult. Further research is needed to more comprehensively evaluate the biomechanical properties of primary teeth and the effects of root canal filling materials on dentin.

Future research should incorporate fatigue testing, nonlinear material modeling, and more complex boundary conditions to better simulate real intraoral environments, thereby enhancing the direct clinical applicability of findings. Furthermore, future studies may be enriched by including a broader range of materials, particularly through the addition of other calcium silicate- and calcium phosphate-based bioceramics, to allow for more comprehensive comparisons. Additionally, evaluating the interaction between root canal filling materials and restorative materials on coronal stress distribution will improve clinical decision-making processes. Finally, validation through in vivo studies or correlation with clinical outcome data is essential to strengthen the clinical relevance of FEA findings. While the current results may provide guidance for material selection in pediatric endodontics, clinical validation remains warranted.

Supplemental Information

Supplemental Information 1 Raw data: MTA solid model under 330 N force

Supplemental Information 2 Raw data: Gutta+Bioroot RCS solid model under 330 N force

Supplemental Information 3 Raw data: biodentine solid model under 330 N force

Supplemental Information 4 Raw data: Gutta+AH PLus solid model under 330 N force

Additional Information and Declarations

Competing Interests

Author Contributions

Clinical Trial Ethics

Data Availability

Clinical Trial Registration

The authors declare there are no competing interests.

Gizem Karagöz Doğan conceived and designed the experiments, performed the experiments, prepared figures and/or tables, authored or reviewed drafts of the article, and approved the final draft.

Yelda Polat Yavuz analyzed the data, prepared figures and/or tables, authored or reviewed drafts of the article, and approved the final draft.

Büşra Karaağaç Eskibağlar analyzed the data, prepared figures and/or tables, authored or reviewed drafts of the article, and approved the final draft.

The following information was supplied relating to ethical approvals (i.e., approving body and any reference numbers):

The University of Fırat granted Ethical approval to carry out the study within its facilities (Ethical Application Ref: 2022/10-14).

The following information was supplied regarding data availability:

The 3D models are available at Morphosource:

– Media 000776214: Mandibular Second Primary Molar And Jaw Model: DOI 10.17602/M2/M776214

- Media 000776202: Mandibular Second Primary Molar: DOI 10.17602/M2/M776202

The following information was supplied regarding Clinical Trial registration:

none.

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
