# Peer review of "Finite element analysis of root canal filling materials in retained primary molars with congenital tooth agenesis"

_PeerJ, doi:10.7717/peerj.20206_

## Round 0.1 · original submission · Major Revisions

Reviewer 1 ·

Basic reporting

The manuscript presents a finite element analysis (FEM) of stress distribution in retained primary molars filled with different canal materials. While the study offers a relevant approach to an understudied area, several aspects could be improved to enhance scientific rigor and clinical relevance.

Experimental design

The study's experimental design could benefit from a broader inclusion of materials. The exclusion of newer or alternative bioactive root-end filling materials, such as calcium-silicate and calcium-phosphate-based bioceramics beyond MTA and Biodentine (e.g., EndoSequence BC Sealer, iRoot SP), limits the comprehensiveness of the analysis. These materials have shown promising bioactivity and mechanical properties and could add meaningful comparison points in future simulations. Additionally, all materials are modeled as linear and isotropic, which does not reflect the complex viscoelastic and anisotropic nature of real dental tissues and materials. Consideration of more realistic material behaviors in future simulations could yield more clinically translatable results.

The loading conditions were limited to static vertical and oblique forces at a single magnitude (330 N). Introducing cyclic loading to mimic mastication over time and varying magnitudes that reflect pediatric bite forces under functional stress could provide a more nuanced understanding of fracture risks. Moreover, the decision to use only one representative tooth limits generalizability. The anatomical variance in primary molars should be accounted for, potentially via analysis of multiple specimens or through parameterized models.

Validity of the findings

The discussion would benefit from deeper exploration of clinical implications, especially regarding how the FEM findings translate to long-term survival and failure modes of treated primary teeth in situ. The paper mentions BioRoot RCS as a promising material but does not engage in depth with how its stress distribution compares functionally and biologically with other modern calcium-silicate materials or how these findings might inform restorative decision-making in pediatric patients with agenesis.

Regarding limitations, the paper does identify a few key ones but could be expanded. Notably, the assumption of complete root canal filling without voids or defects and the absence of aging factors like thermal cycling or microleakage should be explicitly addressed. Moreover, the exclusive reliance on mechanical stress without considering potential biological responses to different materials (e.g., bioactivity, remineralization potential) limits the interpretability of results in clinical practice.

The authors could add a paragraph to compare current results with the following paper: Abedi-Amin A, Luzi A, Giovarruscio M, et al. Innovative root-end filling materials based on calcium-silicates and calcium-phosphates. J Mater Sci Mater Med. 2017;28(2):31. doi:10.1007/s10856-017-5847-1

Additional comments

Future studies should explore the integration of fatigue testing, non-linear material modeling, and more complex boundary conditions to better reflect intraoral environments. Exploring the interaction between root canal filling materials and restorative materials on coronal stress distribution would also be beneficial. Furthermore, in vivo validation or correlation with clinical outcome data would significantly strengthen the FEM-based predictions.

English language usage is generally clear, though revision of phrases such as “liquid filtration or bacterial leakage risk” (line 54) for smoother wording, and replacing “some sensitivity has been observed” (line 55) with more specific language, would improve clarity.

Reviewer 2 ·

Basic reporting

Line 41: Primary teeth can persist primarily, please replace primarily with firstly.
Divide the first paragraph of the introduction into two paragraphs.
Figures 1, 2 and 3 are small and have poor quality.
In the literature, finite element analysis is FEA not FEM.

Experimental design

The major concern with this study lies in the discrepancy between the stated objective and the conclusions drawn. The authors claim that the aim of the study is "to investigate the effect of different root canal filling materials used during endodontic treatment on the fracture resistance of the root of the mandibular second primary molar by analyzing stress distributions and failure risk through FEM models." However, the conclusion merely reports stress distribution results: "The highest von Mises stress was measured in the GP + AH group, followed by MTA, Biodentine, and GP + BioRoot RCS groups, with stress concentrated in the pericervical area of the models."

It is important to note that stress distribution alone does not equate to fracture resistance, unless the study defines clear failure criteria or establishes a correlation with actual fracture data—either experimentally or through validated failure analysis within the FEA framework.

Moreover, I do not believe that finite element analysis (FEA) alone is sufficient to support conclusions about fracture resistance. In the absence of complementary in vitro testing, the findings lack validation, and therefore, in my view, the study should be rejected.

Validity of the findings

As the study design is not adequate, the findings are not real.

Additional comments

The conclusion is not supported by the results, and are too vague.

·

Basic reporting

This manuscript examines how various types of canal filling materials affect tooth stability during root canal procedures for primary teeth with absent permanent molars by performing finite element analysis. This study investigates mechanical behavior of four root canal filling materials through 3D Finite Element Modeling (FEM) according to "Comparison of current canal filling materials used in root canal treatments of primary teeth with congenitally missing permanent tooth germ: a finite element analysis". An enhanced simulation study evaluates stress distribution patterns from occlusal loads exerted on mandibular second primary molars filled with MTA, Biodentine, GP + AH Plus, and GP + BioRoot RCS materials. This research investigates appropriate materials which maintain retained primary teeth structural integrity when treating cases of congenital agenesis. The analysis approach yields reliable results while exploring a new concept in child endodontic treatment. However, the addressing the below comments serves to enhance the manuscript's content clarity and structural organization while maintaining methodological rigor suitable for publication.
1. Title
• The title describes the content well although it remains too lengthy and difficult to understand. Suggest shortening for impact. For example: “Finite Element Analysis of Root Canal Filling Materials in Retained Primary Molars with Congenital Agenesis”
• 2. Abstract
• The abstract lacks clarity in defining the novelty. Clearly state the novelty of this being the first FEM study focused on primary teeth in the abstract.
• Grammatical issues: Rewrite: "is to investigate" to "was to investigate" (past tense indicates completion of study).

3. Introduction
• Additional clarification about knowledge gap should be included in the background section. Highlight how FEM performs best for pediatric endodontics and explain what problems in clinical practice it solves.
• Literature Gap: While prior studies are cited, but insufficient discussion of the gap that this research specifically addresses. Expand the discussion at lines 57–86 to illustrate the lack of biomechanical data on primary teeth treated with these materials.

Experimental design

Materials and Methods
• The study uses 330 N for occlusal loading. While supported by references, it would benefit from citing age-specific bite forces (e.g., Owais et al. and Abu Alhaija et al.) earlier in the section
• Clearer methods need to be displayed for both boundary condition applications and force vector utilization. Example: Researchers applied a total occlusal surface distribution of 330 N force at three distinct points. specify exact location and rationale.
• State more clearly the rationale for assuming isotropic properties in materials such as gutta-percha and dentin, which are known to behave anisotropically. The process behind assuming isotropic and homogeneous properties for individual materials should receive detailed clarification.
• The study fails to justify its selection of static and linear analysis instead of nonlinear or dynamic approaches that demonstrate better physiological loading simulation.
• The methodology must demonstrate mesh independence validation through results which explain mesh density effects or provide an explicit explanation.
• The results show a lack of both sensitivity analyses and reproducibility measurements. Did investigators perform multiple simulations in order to validate mesh independence?

Validity of the findings

• The clinical implication of a ~30 MPa difference between AH Plus and BioRoot RCS in dentin stress should be elaborated. Is this difference significant enough to change clinical practice?

Additional comments

Discussion
• Given the in-silico nature of the study the statement that BioRoot RCS is "more promising" stands strong. Use the wording " within the limitations of FEM modeling ….”to qualify these statements.
• BioRoot RCS demonstrates superior performance without enough explanation of how it achieves better outcomes beyond modulus of elasticity measurements. Provide better interpretation.
• Discuss clinical implications, e.g., could this effect material selection for long-term retention?
• The report mentions the “monoblock effect” without explaining its meaning to readers. For readers without endodontic knowledge the author should provide clear explanation.

Conclusion
Clearly restates key findings. Ends abruptly and without clinical guidance. Include 1–2 sentences on potential future research or clinical recommendations. Add a sentence like: "These findings may guide material selection in pediatric endodontics, but clinical validation is warranted."

Figures and Tables
Figures 1–4: Captions are too generic (e.g., “Mesh_representation_of_3D_solid_model”). Improve to reflect what’s illustrated. Include captions that are descriptive enough to stand alone.

---

## Round 0.2 · Major Revisions

Reviewer 2 emphasized that the authors have addressed all his/her previous comments; however, he/she remains unconvinced by the methodology employed in this study. Thus, the authors need to address these concerns in detail.

Reviewer 1 ·

Basic reporting

The authors have provided all the required improvements.

Experimental design

-

Validity of the findings

-

Reviewer 2 ·

Basic reporting

The authors have addressed all of my comments.

Experimental design

-

Validity of the findings

-

·

Basic reporting

-

Experimental design

-

Validity of the findings

-

Additional comments

The authors have addressed all the comments and suggestions, and the manuscript has dramatically improved. I would like to congratulate the authors and wish them all the very best in their future endeavours.

---

## Round 0.3 · accepted · Accept

Congratulations on your valuable contribution to our journal.

Reviewer 1 ·

Basic reporting

The authors have provided all the required improvements

Experimental design

Ok

Validity of the findings

Ok

Additional comments

None

·

Basic reporting

None

Experimental design

None

Validity of the findings

None

Additional comments

None